# Deep Eutectic Systems as Novel Vehicles for Assisting Drug Transdermal Delivery

**DOI:** 10.3390/pharmaceutics14112265

**Published:** 2022-10-23

**Authors:** Jinbao Wang, Mingjian Li, Langhuan Duan, Yameng Lin, Xiuming Cui, Ye Yang, Chengxiao Wang

**Affiliations:** 1School of Life Science and Technology, Kunming University of Science and Technology, Kunming 650500, China; 2State Key Laboratory of Geological Processes and Mineral Resources, China University of Geosciences, Wuhan 430074, China

**Keywords:** deep eutectic solvent, transdermal drug delivery system, penetration enhancer, solubility, permeability

## Abstract

In recent years, deep eutectic systems (DES) emerged as novel vehicles for facilitating the transdermal delivery of various drugs, including polysaccharides, proteins, insulin, vaccine, nanoparticles, and herb extracts. The objective of this study is to conduct a comprehensive review of the application of DES to transdermal drug delivery, based on previous work and the reported references. Following a brief overview, the roles of DES in TDDS, the modes of action, as well as the structure–activity relationship of DES are discussed. Particularly, the skin permeation of active macromolecules and rigid nanoparticles, which are the defining characteristics of DES, are extensively discussed. The objective is to provide a comprehensive understanding of the current investigation and development of DES-based transdermal delivery systems, as well as a framework for the construction of novel DES-TDDS in the future.

## 1. Introduction

The transdermal drug delivery system (TDDS) became the third most common route of drug delivery after injection and oral administration over the past few decades. However, the development of effective TDDS for a wide range of active pharmaceutical ingredients (APIs) remains a significant challenge. Approximately 40% of available oral drugs and 90% of new chemical entities are reported to have poor solubility and permeability, rendering them ineffective for delivery by transdermal and topical systems due to the impermeable nature of skin [1,2]. To overcome these obstacles, various strategies, including physical methods, pharmaceutical methods, and chemical methods, were developed. Chemical permeation enhancers (CPEs), such as azone, sulfoxide, terpenoids, glycosides, and ethanol appeared to be one of the most commonly employed methods [3].

The following are common ways in which CPEs can improve the skin permeation of APIs: (1) modify the fluidity of the stratum corneum (SC) by disorganizing the lipid alkyl chain [4]; (2) enhance the drug’s skin distribution coefficient [5]; (3) establish a drug reservoir in the SC by generating hydrophilic pores [6]. However, only a small number of CPEs are utilized in commercial products due to the lack of clarity regarding their toxicity and other safety concerns [7]. Therefore, novel biocompatible and biodegradable skin penetration enhancers are urgently required.

Recently, several novel CPEs, including amino acid derivatives [8], alcohols, fatty acids [9,10,11], and esters [12] were developed to enhance the skin permeability of API with minimal irritation or toxicity. Among them, DES is considered a “green” alternative to harmful organic solvents and can be used as CPEs, surfactants, solubilizing/stabilizing agents, and drug reservoirs [13,14]. In particular, the development of biocompatible DES using moieties, such as fatty acids, amino acids, glucose, carboxylic acids, and choline received considerable attention [15,16].

A comprehensive review was conducted in this work to introduce the types, characteristics, and applications of DES in TDDS. The investigation of the permeation mechanism and the structure–activity relationship received special attention. Furthermore, some recent achievements, such as the penetration of macromolecules and nanoparticles (NPs) by DES, received special attention. As a result, this review served as a guide for the design and development of novel DES in TDDS.

## 2. Deep Eutectic Solvent (DES)

### 2.1. The General Rule of DES

DES is defined as a eutectic mixture composed of a hydrogen bond acceptor (HBA) and a hydrogen bond donor (HBD) that results in charge delocalization and a subsequent decrease in melting point [17]. Choline chloride, zinc chloride, and tetrabutylammonium bromide are commonly used as HBA, while urea, citric acid, and oxalic acid can serve as HBD. In general, the stronger hydrogen bonds between HBD and HBA contribute to the formation of DES, while sterically hindered groups can prevent the formation of hydrogen bonds and DES [18]. DES exhibited the same physicochemical properties as ionic liquids (ILs), including low vapor pressure, inflammability, high tunability, and conductivity. Additionally, DES is more likely to be implemented on a larger scale than ILs due to its unique advantages, including ease of preparation, low cost, and minimal environmental impact [19,20]. 

There is no significant difference between DES and ILs in a broad sense. In the reports from Justin, Pavimol, and Kevin [21,22,23,24], DES is typically classified as a similar system to ILs. However, there were still differences in terms of preparation methods and compositions. For instance, cation formation and anion exchange are the major synthesis processes for ILs, whereas there are three primary synthesis processes for DES: heating, grinding, and freeze-drying [25,26]. The majority of intermolecular forces were due to electrovalent bonds. In contrast, DESs were composed of two or three feasible components, with hydrogen bonds accounting for the majority of intermolecular interactions [17,27,28]. 

Certain systems, in particular, choline and its derivatives, can be classified as both ILs and DES [6,29,30]. In this review, the system prepared by the heating method as DES was classified, where the intermolecular interaction was a hydrogen bond rather than an electrovalent bond. Furthermore, some Choline IL was also classified as DES.

### 2.2. Natural Deep Eutectic Solvent (NADES)

In general, the HBA and HBD of NADES are primarily composed of natural molecules, such as amino acids, and primary metabolites, such as organic acids, or sugars. The NADES could be described as a mixture of at least two naturally occurring solid molecules that have a high melting point individually but become liquids when combined at a particular composition [31]. Because of the food-grade nature of their compounds, this type of DES is utilized in a variety of applications, including extraction [32,33], enhancement of bioactivity [34], stabilization [35], or improved aqueous phase solubility [36]. In light of the aforementioned, NADES is considered suitable for “green” solvents to meet the current challenges associated with the search for new pharmaceutical excipients and produce more potent drugs. NADES, for example, is specially designed for the development of wound dressing hydrogels with multiple functions, such as biocompatibility, antibacterial properties, and wound healing ability [37]. The composition and application of some NADES are shown in Table 1.

### 2.3. Therapeutic Deep Eutectic Solvent (THEDES)

THEDES is a system consisting of two or more compounds, one of which is an API. [43,44]. The API can be served both as HBA and HBD. For example, aspirin can be used as the HBD to form DES with choline chloride [45]; while lidocaine and atropine can be used as the HBAs to construct DES with carboxylic acid [46]. In addition to the inherent properties of low vapor pressure, high tunability, or conductivity, this class of DES exhibited superior solubility, permeability, and absorption of the API, resulting in greater biocompatibility/biodegradability than the original form. Consequently, these new eutectic systems provide a sustainable alternative for several pharmaceutical and biomedical applications. A common example is THEDES, a mixture of menthol and camphor that is widely used as a drug solvent with high permeation enhancement. In a reported work [43], a mixture of IBF-L-menthol or IBF-LD-menthol resulted in 2 and 3-fold higher membrane penetration than the corresponding ibuprofen (IBF) powder. The composition and application of THDES are shown in Table 2.

## 3. Role of DES in TDDS

### 3.1. DES as Stabilization Agents

The stability of APIs is essential to their therapeutic effect [51], particularly for biom-acromolecule drugs such as peptides and proteins, which degrade rapidly in vitro and in vivo. The stability of API is related to temperature, pH, humidity, light, etc. [52]. Stability is also an important factor to consider in TDDS formulations, such as patches [23] and emulsions [53], as poor compatibility with the matrix results in crystallization and precipitation of the API, which can have negative effects on the therapeutic activity. Because of its broad compatibility, DES can be used as a stabilization agent in TDDS to improve the stability of both small API and macromolecular substances [54,55,56]. In previously published studies [57], the stabilities of aspirin, clavulanic acid (CLV), and imipenem (IMP) in DES were increased by 8.2, 2.5, and 7 folds, respectively. In addition, DES is advantageous for bio-macromolecular stabilization. DES (choline di-hydrogen citrate-xylitol 2:1) was used to significantly increase the stability of laccase. Thus, the enzymatic activity increased by up to 200% [58]. Mechanism analysis revealed that the HBA/HBD (betaine-xylitol) of DES could protect laccase from denaturation and aggregation triggered by the supramolecular hydrogen bond interactions of NADES [59]. The CAGE-based DES could protect insulin’s α-spiral secondary structure from degradation. Furthermore, as a stabilizing agent, the DES could moderate the drug’s compatibility with the patch matrix. A THEDES based on rotigotine and lactic acid was incorporated into a patch, which demonstrated satisfactory stability without crystallization after six months [60].

### 3.2. DESs as Surfactants

The therapeutic utilization of conventional surfactants, such as SDS, CTAB, SDBS, and TX-100 was problematic due to their inherent toxicity and detrimental effects on eco-systems [61]. DESs with long hydrophobic tails generate numerous types of molecular assemblies, such as microemulsions and micelles, because they typically exhibit surfactant-like activity. Due to their unique properties, amphiphilic DES are promising tools for the construction of self-assembling systems [62]. In general, the weak interaction between surfactant aggregates and precursor ions results in an unstable state. In contrast, the strong effects of hydrogen bonds of DES form a nanostructure, which not only increases the drug’s solubility, but also its ability to penetrate cells [63,64]. Furthermore, the nanostructure was enforced by the longer alkyl chain [65]. In terms of particle size and size distribution, stability, and skin permeability, DES-assisted TDDS is superior to conventional surfactant-assisted TDDS due to the reduced interfacial tension of the DES and API particles [66]. A THEDES composed of lauric acid and lidocaine was employed as a novel surfactant in the development of ME and hydrogel [67]. Choline-oleic acid and [Chl]-[FAs] were utilized as surfactants in micelles for the transdermal delivery of PTX and insulin [68,69].

### 3.3. DESs as Solvents

The partition coefficient (PC) of the SC primarily reflects the drug concentration in the SC and is associated with percutaneous penetration ability through the SC [70,71]. As a solvent, DES could increase the solubility of the API, thereby enhancing the PC in the SC. In addition, it can serve as a reservoir for controlled and prolonged drug release. Numerous studies were conducted on the solubilization effects of DES [36,72,73]. By changing the type of HBA, DES could significantly increase the solubility of both hydrophilic and hydrophobic API. For instance, a complex composed of curcumin, choline, and oleic acid DES substantially increased the drug’s solubility from 30 nmol to nearly 22 mmol [74]. Choline-amino acid DES increased the solubility of paclitaxel by a factor of 10,000 [75]. When oxalic acid and tartaric acid were used as HBA, the solubility of narcotics, such as lidocaine and procaine, was enhanced [72,73,76]. In addition, CAGE could increase the solubility of sorafenib to 500 mg/mL, leading to improved bioavailability with a half-life of 2 and a plasma peak concentration of 2.2 [77].

The enhancements of SC were attributed to the solvation effects of DES, which could eliminate the self-association of the solvent and decrease the interfacial tension between API and the solvent [78]. Furthermore, DES and API can interact via hydrogen bonds, der Waals forces, and π-π bonds [79,80]. The solubilization effects of DES depend on their structural properties, including their electron distribution, alkyl side chain, number of alkyl groups, and HBA/HBD type. In a reported work [81], L-alanine, L-proline, and L-methionine were employed as HBD to form DES with choline chloride. However, only choline chloride-proline solubilized zafirlukast (ZFL), which may be attributed to the H-bond between the aromatic ring in ZFL and the carbamate in proline. ChCl-P exhibited the highest solubility to acetaminophen in comparison to choline–malic acid (ChCl-MA) and choline–tartaric acid (ChCl-TA), which can be attributed to the dominance of strong hydrophilic ion interactions and hydrophilic interactions among the constituents of ACP and NADES, as described in previous work [82].

## 4. Influences of DES on SC

The DES permeation mechanism is critical for understanding the mechanisms of action and designing novel CPEs in TDDS. The “Brick-mortar” structure of SC is one of the most important models for understanding the permeation mechanism of DES. The general mechanisms of action of DES include liquid modification, protein modification, and liquid exchange. In a previously published study [83], 31 DESs were evaluated; 70% exhibited lipid extraction functions and 30% suggested lipid fluidization effects. In our reported work [16,84], several amino acid DESs were synthesized, and 62.5% of the total exhibited the diverse effects of fluidization and liquid extraction.

### 4.1. Liquid Modification

DES can change the structure of the skin to make it more permeable by disrupting the keratinocytes’ regular and tight arrangement [85]. It was reported that hydrophilic DES could open the tight junctions of SCs and improve the fluidization of the lipid region, thus facilitating the intercellular transport of API. On the other hand, hydrophobic DES can increase transport channels for the transcellular delivery of API [86]. It was demonstrated that amphiphilic DES could enter a lipid bilayer and close the hydrophobic edge of the lipid bilayer, eventually destroying the lipid membrane [87]. Previous studies show that the HBA of DES can intercalate SC, disrupting lipid bilayer accumulation within tissues, whereas HBD increases SC hydration capacity [88]. Further investigation revealed that the long carbon chain on the amino acid HBA is capable of transforming the lipid of SC into a liquid crystal state with enhanced fluidity. Therefore, previous research demonstrated that DES can play a significant role in the liquid modification of SC [6].

### 4.2. Keratins Modification

Keratins in SC are responsible for maintaining the hexagonal structure of keratinocytes, thereby contributing to SC’s toughness and flexibility. DES can interact with keratin in keratinocytes, causing protein conformation changes and damaging keratinocyte arrangement, or denature keratin to form vacuoles, thereby facilitating drug penetration [89]. Simulations of molecular dynamics demonstrated that DES forms strong interactions with proteins, which could replace the relatively weak protein–water hydrogen bonds and lead to the formation of more stable salt bridges [90]. Zhang et al. demonstrated that DES can form holes in keratins, thereby accelerating drug delivery [91]. Qi believes that choline DES decreases the efficacy of the SC barrier by disrupting the keratins’ ordered arrangement [6].

### 4.3. Lipid Exchange

In comparison to traditional CPE, DES exhibits significant penetration-promoting activity for macromolecular drugs when used as drug solvent. This new mechanism of action is attributed to the “lipid exchange” effect, in which DES selectively dissolves and replaces the lipids, allowing them to diffuse into the deeper skin layers. Hansen solubility parameters (HSP) revealed that the choline and amino acid-based DES had HSP values that were closer to those of the skin, indicating greater affinity and skin permeability [16]. Therefore, DES can completely infiltrate the dermis and form a surface-saturated layer, which then diffuses into deeper layers under the force of concentration gradient and replaces the lipid components in the skin [6].

## 5. Structure–Activity Relationship

### 5.1. Compositions

In DES, the HBA, such as choline [30,92,93], amino acid [94,95,96], menthol, and thymol [97,98], are the key factors that dominate their skin permeability, while the HBD can moderate/regulate the properties. In previous research, a series of amino-based DESs were synthesized; when the HBD was fixed, the skin permeability of DES could vary depending on the amino acid species [84]. In Wu’s study [2], ibuprofen was used as HBA, while other drugs (tetrahexylammonium bromide, procaine, lidocaine, etc.) were combined as HBD to form nine different DES. The system containing tetrahexylammonium bromide and ibuprofen provided the most effective penetration. In addition, Samir Mitragotri and co-workers [29] prepared a series of DES containing choline as HBA and nine different organic acids as HBD, each with a different number of carbon atoms, logP, and pKA, among other properties. The DESs demonstrated significantly different promotion effects toward the model drugs after skin application, with permeation amounts ranging from 42 to 108 [µg/cm^−2^], which were attributed to HBD properties.

In reported works [30], the chemical shifts of hydroxyl protons in DES were analyzed by ^1^H-NMR spectroscopy and correlated with the permeation enhancement ratio (ER). In terms of skin permeability, it was discovered that the interaction between HBA and HBD exhibited “competitive inhibition”. This interaction affects the melting point, viscosity, and stability of DES. If an excessive number of intramolecular H-bonds are formed, the number of intermolecular hydrogen-binding sites between DES and drugs are drastically reduced, which has an adverse effect on drug solubility and skin permeability. Nuclear Overhauser effect spectroscopy (NOESY) provided additional confirmation of the theory: the number of cross peaks of cation and anion ions is negatively correlated with the permeation effect. Therefore, reducing the intramolecular interaction in a certain range can enhance DES’s ability to promote permeability [30].

### 5.2. HBA/HBD Ratio

The ratio of HBD to HBA in the formation of DES is also essential to its performance. However, there is not yet a universal agreement on how the molar ratio works. In a published study, ILs were constructed using choline bicarbonate and vanillic acid in the ratios 1:4, 1:2, 1:1, and 2:1 [30]. In terms of transporting insulin to the dermis, it was discovered that a ratio of 1:2 was superior to other ratios. Similarly, the DES composed of choline and geranic acid demonstrated the greatest skin permeability at a ratio of 1:2. However, in a menthol (MN)-capric acid (CA) DES system, a 7:3 molar ratio produced the best solubility and skin permeability results for risperidone (RIS) [99]. In a paeonol-matrine system, the optimal molar ratio for paeonol skin permeation was determined to be 3:7 [100]. 

The changes in molar ratio first alter the viscosity [101], conductivity, and solubility [102] of DES, thus affecting the diffusion rates of drugs. In addition, DESs with a higher HBA ratio exhibited improved extract lipid extraction and fluidization abilities. Furthermore, NOESY revealed that the HBA/HBD ratio influenced the intramolecular/intermolecular interaction balance. For instance, in choline DES, when the proportion of choline was high, the intramolecular action of the methyl group bonded to the nitrogen atom predominated; however, when the proportion of HBD was high, the intermolecular bond became the dominant interaction [2].

### 5.3. Substituents

The performance of such DES or surfactants, including the solubilization effect, micelle behavior, hydrophilic and lipophilic balance characteristics, etc., is significantly affected by the substituent on HBA, especially the length and position of hydrophobic chains. This is essential for DES, as its mechanism increases drug solubility and distribution in the SC.

The HBA with a symmetric structure and three carbon atoms was more effective at promoting penetration in choline DES [29]. The quantitative structure–activity relationship (QSAR) analysis of our earlier works [88] revealed that the electrochemical properties were the main factor in the permeation enhancement. It was discovered that the change in carbon chain length on the HBA had a greater impact on the skin permeation of drugs than the change in the anion. There was a positive correlation between the length of the carbon chain and the ER value.

## 6. Permeation Effect

### 6.1. Small Compounds

Nonsteroidal anti-inflammatory drugs (NSAIDs) administered topically are a valuable and effective alternative to oral administration. However, the majority of these drugs have poor water solubility, limiting their use in hydrophilic drug delivery systems, such as patches [103,104]. In a reported work [103], a hydrophilic arginine-glycerol DES system with a molar ratio of 1:4 was prepared as a solvent for ibuprofen. Compared with water, the solubility of ibuprofen increased to ~8000-fold in the DES. As a result, the drug’s stability was significantly improved, and its permeability was increased by up to 8.5 times. In another work [91], lidocaine-ibuprofen-based THEDES was developed as the solvent for the antimalarial drug artemisinin. Compared to commonly used vehicles, such as PEG 400 and IPM, the skin permeation flux of artemisinin was significantly increased in the DES, which could be attributed to the high capacity of the DES to dissolve ARS. Furthermore, DES can improve not only hydrophobic but also hydrophilic drugs. The reported work used acarbose (logP = 6.8) and ruxolitinib (logP = 2.9) as model drugs, and the choline-geranic acid (CAGE) DES demonstrated significant improvements in both intradermal and transdermal delivery of the two drugs [29].

### 6.2. Biomacromolecule

In contrast to lipophilic drugs, the transdermal delivery of large biomacromolecules is complicated by the strong barrier properties of the skin, particularly the SC, which is the outermost layer. Due to the tightly packed lipid-filled intercellular space surrounding the keratin and water-based intracellular environment, the SC hardly permits the permeation of macromolecules in its undisturbed state [105,106]. To successfully penetrate the skin, many bioactive macromolecules, including insulin, proteins, RNAs, and DNAs, find it difficult to pass through the natural SC [107]. In recent years, the DES system, particularly the choline derivatives DES, demonstrated overwhelming superiority in improving the skin permeability of biomacromolecules [83,108]. The applications of DES for the transdermal delivery of biomacromolecules are shown in Table 3.

#### 6.2.1. Insulin

Transdermal delivery of insulin remains difficult due to the ineffectiveness of conventional CPEs; however, the DES demonstrated significant skin permeation enhancements on this biomacromolecule without changing its secondary structure. Mitragotri and his co-workers were the first to report the stimulatory effects of DES containing choline [30]. In their study, choline was the HBA and geranic acid was the HBD. The formed CAGE-based DES may facilitate insulin penetration into the epidermis and dermis through the SC. According to quantitative analysis using the Franz diffusion cell assay, it was found that the amount of insulin that penetrated the dermis of porcine skin at 12 h was ~5.5 µg cm^−2^ and at 48 h, it was ~17 µg cm^−2^. Following that, the choline-oleate composition demonstrated similar activities, with oleate serving as the HBD [92]. In addition, it was discovered that these DES exhibited a composition-dependent permeation mode, with formulations containing an excess of H-donor exhibiting superior performance [108].

#### 6.2.2. Protein

The CAGE DES also improved skin permeation of proteins such as bovine serum albumin (BSA) (~66 KDa) and ovalbumin (OVA) (~45 kDa), which are both inhibited by the SC when administered via skin [108]. The Franz diffusion cell assay revealed that CAGE significantly increased BSA penetration through the deep layers of porcine skin in a time-dependent manner. The protein delivery to the dermis (~1.6 µg cm^−2^) and epidermis (~2.8 µg cm^−2^) 12 h after administration of the DES-BSA formulation was significantly higher than that of the PBS controls (dermis: ~0.04 µg cm^−2^ and epidermis: ~0.08 µg cm^−2^). In addition, the amount of BSA in the receptor cell was ~6 µg cm^−2^ at 48 h, whereas BSA in PBS penetrated the skin poorly at all time points.

#### 6.2.3. Polysaccharide

Polysaccharides are a class of naturally occurring polymers composed of glucose chains connected by glucoside bonds. Wu and co-workers [109] designed a novel DES using malic acid and choline to introduce the hydrophilic polysaccharide dextran into the skin. Both in vitro and in vivo skin penetration studies indicated that the choline-malic acid DES could significantly increase dextran penetration while causing negligible irritation to mice’s skin and toxicity to human epidermal cells. Another investigation using a variety of dextran with different MWs revealed that the DES of CAGE (1:2) provides the most efficient transdermal transport of Dextran up to 150 kDa [83].

#### 6.2.4. siRNA

Small interfering RNA (siRNA) cannot be delivered into the skin effectively due to the skin’s poor stability and barrier function. Mitragotri and his co-workers successfully delivered the NFKBIZ siRNA into the dermis using the CAGE + CAPA combination as the mixture solvent system [110]. Through molecular simulation, it was shown that the DES system, in particular, improved siRNA stability and provided DES-induced solvating and intercalation effects. This was probably due to the addition of phenylpropanoic acid to CAGE, which also increased relative molecular mobility or decreased local viscosity and improved interactions and compactness between the optimized IL system and the RNA. In addition, the reduced viscosity of the DES system may reduce the intramolecular stress placed on RNA. Psoriasis-related signals, such as TNF-α and IL-17A, were downregulated due to the suppression of aberrant gene expression caused by DES-siRNA skin treatment.

#### 6.2.5. Antigen Peptide

The skin permeability and solubility of a water-soluble antigen peptide were mediated by choline fatty acid DES, which was developed as biocompatible CPEs in a previously published study [111]. Due to its low cytotoxicity and ability to mediate skin permeability, oleic acid (C18:1) was chosen from the candidate HBD. The peptide’s transdermal delivery flux increased 28-fold when compared to that in an aqueous vehicle. Furthermore, IL-mediated transcutaneous vaccination inhibited tumor growth in vivo more effectively than injection. Furthermore, Shihab Uddin and co-worker [112] created a transdermal delivery system for leuprolide acetate, a model hydrophilic peptide, based on EDMPC (1,2-dimyristoyl-*sn*-glycerol-3-ethyl-phosphatidylcholine)-C_18_ fatty acids DES. Compared to an aqueous carrier, this novel DES carrier increased the peptide’s skin permeability 65-fold. Chowdhury et al. [113] developed a DES-assisted delivery system for the co-delivery of OVA in transcutaneous immunotherapy. The incorporation of DESs into the drug carrier was found to significantly improve the skin permeability of OVA [108].

**Table 3 pharmaceutics-14-02265-t003:** The applications of some DESs for transdermal delivery of biomacromolecules.

API	HBA	HBD	Ratio	Application	References
Insulin	Choline Choline	Geranic acid Geranic acid	1:4/1:2/1:1/2:1 1:1/1:2	Provided a slow, consistent lowering of the blood glucose level over 12 h Promoted transdermal penetration of insulin	[30] [92]
	Choline	Oleic acid			
BSA, OVA, Insulin	Choline	Geranic acid	1:2	Increased the ability of protein to penetrate the skin and lowed blood glucose	[108]
Dextran	Choline	Malic acid	1:2	Enhanced transdermal capacity and penetration	[109]
	Choline	Geranic acid	1:2	Promoted transdermal penetration of macromolecules up to 150 kda	[83]
siRNA	Choline	Geranic acid	1:2	Effective delivered of siRNA into the skin and suppress aberrant gene expression	[110,114]
		Isovaleric acid			
		Phenylpropionic acid			
		4-Phenolsulfonic acid			
		Phenyl Phosphate			
		Biphenyl-3-carboxylic acid			
IFN-α2	Choline chloride	Fructose	1:1	Provided environmental protection and enhanced stability	[115]
		Citric acid			
		Malic acid			
RA-XII	Betaine	DL-Mandelic acid	1:1	Increased stability and permeability	[116]

### 6.3. Rigid Nanoparticles

The rigid and solid nanomaterials, such as mesoporous silica nanoparticles (MSNs), are incapable of penetrating the compact and regular SC structure, in contrast to the lipid nanovehicle, which has a high skin affinity and provides easy access to the dermis. It was generally accepted that rigid NPs are more likely to be retarded by the superficial layers of SC than to penetrate skin [117]. One of the most significant challenges is instability. Particle aggregation and sedimentation were caused by the nanosystem’s enormous specific surface area and free energy. The other barrier is the transdermal delivery characteristics of the NPs. In contrast to the solute–solvent system, the rigid NPs were completely insoluble in weakly interacting mediums. The mediums were unable to “pull” the NPs into the skin despite the high skin permeability. Therefore, transdermal diffusion between the NPs and the mediums is not synchronized, and the accumulation of abundant NPs on the skin surface is slowed. Thus, the only effective methods for applying rigid NPs to the skin are invasive procedures, such as subcutaneous injection and microneedle administrations [118,119].

In our reported work, an amino acid–citric acid (AACA) DES system was previously developed for the noninvasive delivery of rigid MSNs via skin application [16], as shown in Figure 1. The DES–MSN system was developed by modifying the surface of MSNs with HBD (CA) and heating them with HBA (AA). The MSNs were captured and prevented from sedimentation by a network formed by the strong intramolecular hydrogen bonds between AA and CA. Additionally, the NPs were dispersed widely due to the covalent bonding between the MSNs and the surrounding DES. The strong interactions between the NPs and the DES during skin application allowed the DES to “drag” the nanoparticles across the SC and into the deeper layers of skin. This method allowed rigid NPs to be delivered transdermally. The MSNs NPs were able to access blood circulation through skin application for the first time using the DES strategy.

The skin penetrability was attributed to so-called “Drag” effects from the DES system, which were primarily depicted in two aspects: the high skin of DES permeability and the intense interaction between the NPs and DES. High skin permeability of the DES was accompanied by “erosion” of the SC resulting from the effects of extraction and lipid fluidization [84,120]. In contrast, the DES provided strong interactions between the DES and the NPs, allowing the NPs to be “anchored” to the system by participating in the development of DES. Strong interactions between DES and NPs resulted in the formation of a cross-linked network that “immobilized” the NPs to the DES [121]. A driving force and “Drag” effects were therefore exerted for the transdermal penetration of MSNs by the DES’s high skin permeability and strong interaction with MSNs.

### 6.4. Complex Herb Extracts

In traditional Chinese herbal medicine, a formulation always consists of three to six species of herbs or herb extracts, each of which contains a unique and complex active ingredient. Unfortunately, very few synergistic and synchronous transdermal delivery strategies for these complex systems are developed [122,123]. Particularly, the herb extract contained both hydrophilic and hydrophobic components, necessitating comprehensive compatibility with the TDDS matrix.

Based on its unique properties, DES appeared to be the ideal solvent for the skin permeation of herbal extracts. We used the Chinese herbal medicine formulation “sanwujiaowan,” which consists primarily of six ingredients, as the model drug for investigation [124] (Figure 2). All six ingredients increased skin permeability synergistically and synchronously when combined with an arginine-citric acid (3:1) DES as a solvent. The six ingredients enhance ratios (ER) were found to vary between 1.6 and 4.8.

## 7. DES-Based TDDS

As previously mentioned, DES show great promise for skin penetration. In addition, their distinctive properties enabled their incorporation into a variety of TDDS, including microemulsions, hydrogels, micelles, and patches.

### 7.1. DES-Based Microemulsions

Microemulsion (ME) is a commonly employed TDDS that consists of an aqueous phase, an oil phase, a surfactant, and a co-surfactant. ME is an excellent TDDS due to its unique properties, which include low surface tension, small droplet size, and high skin affinity. However, as a nanodispersion system, the stability, drug capacity, and biosafety of ME are the most significant concerns. Similar to ILs, DESs are demonstrated to be the ideal substance for preparing ME. Depending on their hydrophilicity or hydrophobicity, the DES could be used as both an aqueous phase and an oil phase [125]. Additionally, some amphiphilic DES can be utilized as co-surfactants and surfactants [67,120,126]. 

Viscosity, pH, surface tension, and other properties can be modified in DES-based ME by adjusting the type and ratio of HBD/HBA. The stability of DES–ME is greater than that of the normal formulation. For instance, the clinical application of curcumin was limited by its poor stability and low solubility. Therefore, a hydrophobic DES composed of tetra-n-butylammonium chloride and n-cecanoic acid (1:2 in mole ratio) was used as the oil phase to create the ME, which demonstrated exceptionally high solubility and significantly enhanced stability (against sunlight and long-term storage) of curcumin [127]. In another work [53,128], a DES choline chloride-glycerol (1:2 in mole ratio) was used as the water phase of the ME to increase the amount of resveratrol entrapped and its skin permeability. Similarly, a choline-oleic acid-based ME exhibited solubilization effects on a number of API, including celecoxib, acyclovir, methotrexate, and dantrolene sodium, respectively [129].

It is interesting to note that DES can be prepared from the drug itself. According to early reports, a DES system was developed using the combination of paeonol and menthol [130]. As a result, the THEDES was used directly as the oil phase. The THEDES microemulsion demonstrated good physical stability with the smallest droplet size and lowest viscosity. The permeation flux of paeonol in this formulation was therefore the highest. Others developed the lidocaine-based DES-ME. For example, lidocaine-prilocaine (1:1) [131] and lidocaine-ibuprofen (1:1) [91] were prepared and served as the oil phase in ME, respectively. In contrast, lidocaine-lauric acid (1:1, 4:6) was an amphiphilic DES that can be utilized as surfactants/co-surfactants in ME [67]. The drug-based DES-ME exhibited high skin penetration profiles, which could be ascribed to the drug’s high drug capacity and decreased melting point upon transformation into DES.

Additionally, the DES-ME was effective for macromolecular applications. Choline–fatty acid ([Chl] [FAs]) DES was reportedly used in a ME formulation for insulin transdermal delivery [69]. In the development of the MEs, the DESs were utilized as surfactants, Span-20 as a cosurfactant, choline-propionate IL as an internal polar phase, and isopropyl myristate as a continuous oil phase. The biological activity of the formulation was maintained at 4 °C for 4 months, indicating that the DES could significantly improve its stability. The ME significantly improved insulin transdermal permeation via the intercellular route after skin application through a fluidity-improving mechanism. In addition, DES ME significantly decreased blood glucose levels (BGLs) in comparison to subcutaneous injection.

In some instances, pH-responsive ME could be produced by incorporating specific DES. Menthol and n-octanoic acid (OA) (1:2) could be assembled into hydrophobic DES with pH-responsive profiles. When the pH of the MEs was changed, a phase transition was observed, and a nanoemulsion was produced based on the pH response of the MEs. Consequently, these characteristics of DES-ME demonstrated promise for controlled and pH-responsive drug delivery [132]. Table 4 and Figure 3 illustrate several DES-microemulsion preparations and their respective effects.

### 7.2. DES-Based Hydrogel

Hydrogel, a type of three-dimensional network structure that is incredibly hydrophilic, can quickly swell in water and retain a significant amount of water without dissolving in this swollen state [135]. Starch [136], hyaluronic acid [137], chitosan [138], carbomer [139], and alginate [103] are the most common copolymers used in hydrogel production.

Recently, DES was successfully incorporated into hydrogel systems for a variety of applications. DES-hydrogels had distinct properties when compared to conventional hydrogels. DES demonstrated enhanced mechanical properties as a result of its strong hydrogen bonding. Additionally, by incorporating the functionalized DES, specific properties, including self-recovery, heat transfer, and conductivity can be achieved [140,141,142]. In our previous work, amino acid-based carbomer hydrogels were created. The good compatibility of carbomer and DES resulted in the formation of a homogeneous network microstructure, whereas the strong interaction between HBD and HBA enforced the mechanical and viscoelastic properties [124].

The use of hydrogel beads made of alginate (ALG) and chitosan (CHI) as vehicles in controlled drug delivery studies is extensively researched. However, the requirement for organic solvents limits the encapsulation of compounds with low water solubility [103]. According to a published study, curcumin was successfully encapsulated in ALG-CHI beads using DES (choline chloride-glycerol, 1:1) as a drug-solubilizing vehicle. This encapsulation strategy holds great promise for the water-insoluble compound curcumin [138]. Similarly, a hydrophilic DES derived from arginine-glycerol (1:4) was formulated and incorporated into alginate gels. The novel DES-hydrogels significantly improved the anti-inflammatory efficacy of drugs [103]. 

Because of their high solubility in a wide range of API, DES-hydrogels are ideal for complex Chinese herb formulations. NADES–hydrogel systems for the synergistic transdermal administration of Chinese herbal medicine and local treatments for rheumatoid arthritis (RA) are reported [124]. To develop the DES-hydrogel, carbomer 940 solution was mixed in a 3:1 ratio with arginine and citric acid before being co-heated. The strong interactions and good compatibility between polymer and DES resulted in a homogeneous microstructure with improved mechanical and viscoelastic performance, which can make topical administration easier by improving skin adhesion and retention. As a result, the DES increased the permeability of the skin, while the hydrogel provided a continuous and sustained permeation profile. This increased the efficiency of percutaneous absorption and the therapeutic value of RA.

The functional DES-hydrogels may also be used as a type of dressing for wounds. Wang and his co-workers described a green method for producing a biodegradable wound dressing hydrogel with Aloe vera as the active ingredient. A natural DES (choline chloride-glucose, 1:2) was utilized as a green solvent due to its effective solubilization and stabilization properties [143]. Furthermore, this group reported a multifunctional hydrogel containing a natural DES (choline chloride-mannose, 1:2) for wound dressing [37]. Both mannose and choline chloride were useful nutritional supplements in this formulation, with specific anti-inflammatory properties for wound healing. The robust adhesion properties of DES allow it to securely cover the wound site and isolate it from external contaminants and foreign objects. In a previous study, choline-glycolate (CG-LY) DES was added to gels loaded with copper ions (Cu^2+^) to promote gel formation via intermolecular hydrogen bonds [144]. Drug-resistant bacteria can be eliminated from the wound by the DES’s ability to stimulate the release of Cu^2+^ and produce hydrogel free radicals (OH). Furthermore, the excellent transdermal ability of CG-LY allows the released Cu^2+^ to promote cell migration and stimulate wound healing. Figure 4 depicts some of the applications of DES-hydrogel.

### 7.3. DES Based Micelles

As shown in Figure 5, DES-assisted self-assembly systems are a promising tool for transdermal drug delivery [145]. When DES was used to form self-assembly systems, it increased surface activity and interfacial adsorption, resulting in a change in nanobehavior, including a decrease in droplet size and zeta potential, but an increase in dispersity and drug solubility [62]. Critical aggregation concentrations (CAC), which are important micelle parameters, were particularly affected. A previous study discovered that the length of the alkyl chain of the HBD was related to the change in the CMC value of the micelles [146]. A micelle formulation containing cholinium-oleate (1:1) DES was developed for the transdermal delivery of PTX [68,147]. The solubility of the drug in the micelles was enhanced by multiple hydrogen bonding and cation−π and π–π interactions between the drug and the DES. Additionally, the DES enhanced physical stability and skin permeability. The sizes and size distributions remained constant throughout the entire storage period, and transdermal evaluations revealed that DES-micelles were four times and six times more effective than tween 80-based and ethanol-based formulations, respectively. Isabelle s. Kurnik et al. synthesized a variety of DES using Choline as the HBA and various carboxylic acids as the HBD [148]. As a DDS, these DES could be used to encapsulate curcumin in nanomicelles. Temperature responsiveness and long-term release profiles were also observed in the micelles.

### 7.4. DES-Based Patches

CAGE as a transport facilitator and chitosan as a mucoadhesive matrix were used to develop biodegradable polymeric patches for buccal insulin delivery due to CAGE’s significant effects on insulin stabilization and skin permeation [24]. Insulin was combined with ILs/DES to produce a viscoelastic CAGE gel, which was then placed between two layers of a biodegradable polymer. According to the rheological analysis, the DES patch exhibited viscoelastic properties similar to those of a solid. The cumulative insulin transport across the ex vivo porcine buccal with CAGE was seven times greater than the control. In vivo testing revealed that serum insulin levels were maintained while BGLs were reduced in a dose-dependent manner.

In a patch formulation, the API could also be transformed into DES [149,150]. Three APIs, including imipramine HCl, ascorbic acid, and catechol, were formulated into DES and prepared into patches using gelatin as the matrix, resulting in enhanced percutaneous absorption profiles [149]. The drug–polymer miscibility and skin permeability are particularly important in this situation. According to Fang and co-workers, when rotigotine was converted into DES with lactic acid (ROT-LA) and loaded into a patch, the strong interaction between DES and matrix inhibited the growth of drug crystals, thereby enhancing skin permeability and drug–polymer miscibility [60]. Figure 6 shows some applications of DES-patches.

### 7.5. Polymerized Drug-Based DESs

Polymerized drug-based DESs emerged as an effective strategy in the development of TDDS for increased skin permeability. It was reported that acrylic acid [151], methacrylic acid [152], as well as 1,8-octanediol [153], were all reported to be polymerization monomers. DESs were formed by the formation of hydrogen bonds between the drug and the monomers. Due to free-radical frontal polymerization, polymerized DESs exhibited a homogeneous, solid monolith without drug segregation, yielding polymer monoliths with a high drug concentration. Furthermore, the drug was released in a controlled manner based on the ionic strength, pH, and solubility in the medium.

## 8. Toxicity of the DES in TDDS

When evaluating the biosafety of DES for use in TDDS, biocompatibility and the potential for causing skin irritation are the most important factors to consider. Some ILs, including imidazolium and pyridinium ILs, were recently discovered to be less safe than previously thought due to cytotoxicity and poor biodegradability [154]. In the case of 1-butyl-3-methylimidazolium hexafluorophosphate ([BMIM] [PF6]), for example, it was observed that the hexafluorophosphate anion can decompose in an aqueous acidic medium, resulting in the formation of 1-butyl-3-methylimidazolium fluoride hydrate and, consequently, the toxic product HF [155,156]. DES, particularly NADES, which are derived from natural resources, appeared to be a significantly safer solvent than ILs. Various models, including fish cell line [157] and human HEK-293 cell line [158] were used to investigate the toxicity of DES and ILs, and the results indicate that DESs were significantly less toxic than ILs, such as imidazolium- and pyridinium-based ILs. Further QSAR analysis revealed that the molar ratio of HBA to HBD and the number of HBD carbons had negative effects on the cytotoxicity of the NADESs, whereas the constitutional descriptors of HBD had a positive effect [159].

The potential for skin irritation was the primary concern regarding TDDS. Irritation of the skin is defined as locally reversible damage to the skin caused by contact with the test substance, typically manifesting as erythema and edema [160]. Some conventional CPEs, such as fatty alcohols (decanol and undecanol) and azone, may cause minor skin irritation, thereby limiting their application. Although the skin toxicity of DESs is still under debate, it us shown that imidazolium ILs with long alkyl chains are more likely to cause swelling and inflammation of the skin [120,161]. In general, the toxicity of DES was closely related to its structure, but it was significantly lower than that of ILs. A thorough investigation was conducted between 31 DESs and 44 conventional CPEs, and the DES demonstrated a significantly lower degree of loss in α helices in keratin, which is known to correlate with a lower potential for skin irritation [6]. Another study examined the histological changes caused by DES in the skin. The results show that DES acted as a transient disruptor of the skin structure, allowing bioactive compounds dissolved in it to pass through without harming the cells, which was consistent with the findings of Sidat [3], Wu [162], and Marei [163]. Previous research examined the skin irritation potential of AACA DES [16,84]. At the application sites, neither 3T3 nor HaCaT cells exhibited obvious edema, inflammatory cell infiltration, tissue necrosis, or toxicity.

## 9. Challenges and Prospect

DESs developed significantly over the past decade and found numerous applications in the pharmaceutical industry due to their unique characteristics of nonflammability, high tunability, low vapor pressure, low toxicity, and thermal and chemical stability. It has the potential to be used as a drug solubilization and stabilization vehicle in TDDS, as well as in the production of DES-based TDDS, such as MEs, hydrogels, micelles, and patches. Most importantly, the DES significantly improved skin permeability. Despite the rapid progress, numerous ongoing challenges significantly impede the practical translation of these newly developed DESs from lab research to pharmaceutical commercialization, particularly in the TDDS area.

First, thermal treatment is the most practical method for producing DESs due to its attractive simplicity [28]. However, research on the effects of the empirical method on product performance is limited. The relationship between process parameters and DES properties is still unknown. For example, the effects of heating time and temperature on DES performance, such as viscosity, adhesion force, and surface tension, are unclear. As a result, more research should be conducted to establish a link between the preparation process and the properties of the product, which is required for the design of DES with the desired functions. Furthermore, a moderate method for these thermosensitive substances must be developed.

Second, in addition to drug delivery applications, THEDESs demonstrated their pharmaceutical activities, such as anti-bacterial, anti-fungal, and anti-cancer activities [44,164]. Despite encouraging in vitro results, more research is needed before clinical applications can be made. In terms of regulatory issues, it is still necessary to ensure that the findings adhere to strict regulations and the standards established by international and governmental drug organizations.

Third, further research is required to reveal the DES permeation mechanism in terms of bio macromolecules and rigid NPs. A “drag effect” hypothesis, in particular, was previously proposed to explain the effect of DES on rigid NP penetration, which requires further investigation and verification [16]. For instance, it is necessary to investigate the mode of combination of DES with NPs, the microstructure of DES-NPs, and the formation mechanism of the system. In addition, it is essential to investigate the penetration characteristics of NPs and to identify the key variables for transdermal delivery in relation to the DES bonding force.

Finally, smart nanovaccines based on the specific properties of the DES system can be developed. As DES provided a noninvasive method for the penetration of rigid NPs, functionalized NPs could be incorporated into DES systems to achieve a controlled and long-lasting transcutaneous immunization. In contrast to the conventional subcutaneous delivery method, DES-mediated percutaneous immunization was performed non-invasively, allowing for the formation of a drug reservoir in the skin and thereby enhancing the benefits of long-term and low toxicity of transcutaneous immunization.

## Figures and Tables

**Figure 1 pharmaceutics-14-02265-f001:**
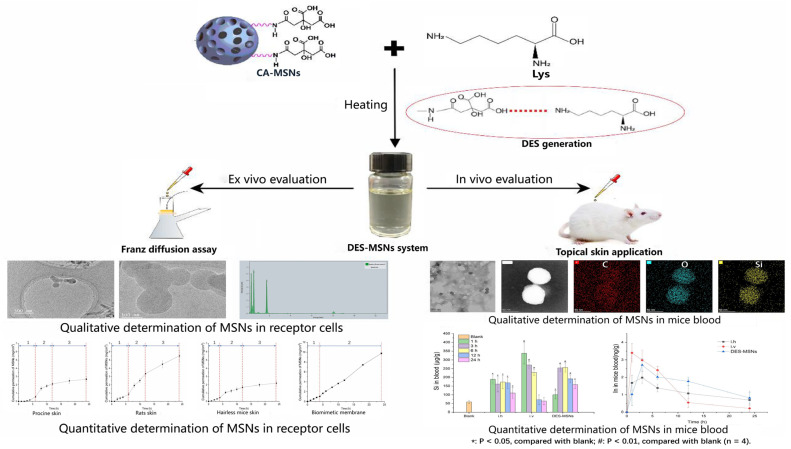
Skin drag effect and permeation-promoting mechanism of the eutectic system on rigid mesoporous silica nanoparticles.

**Figure 2 pharmaceutics-14-02265-f002:**
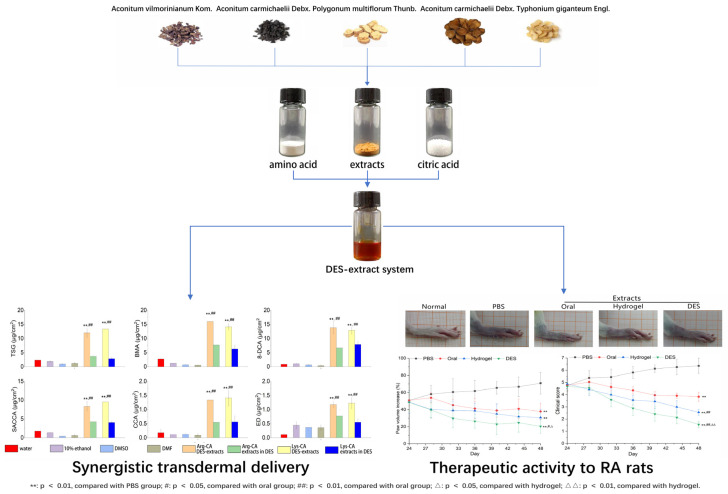
The treatment of arthritis with the active ingredients of “sanwujiaowan” extract through the deep eutectic solvent–hydrogel system.

**Figure 3 pharmaceutics-14-02265-f003:**
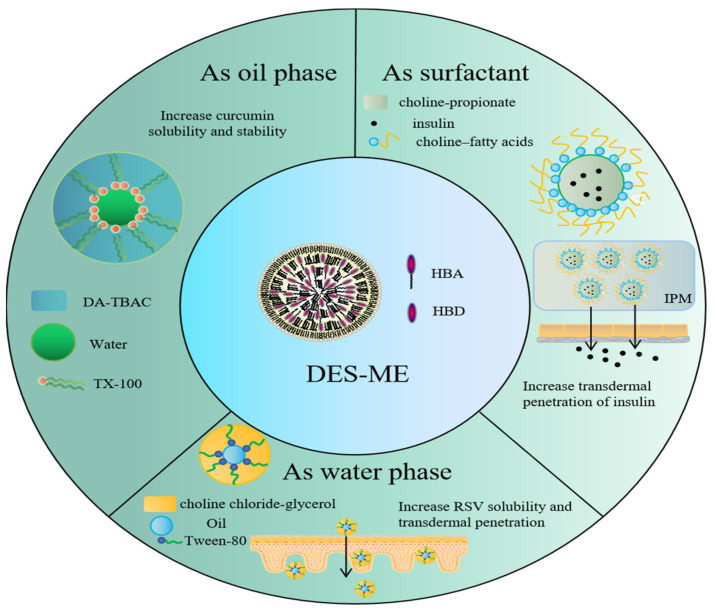
Applications of DES-microemulsion.

**Figure 4 pharmaceutics-14-02265-f004:**
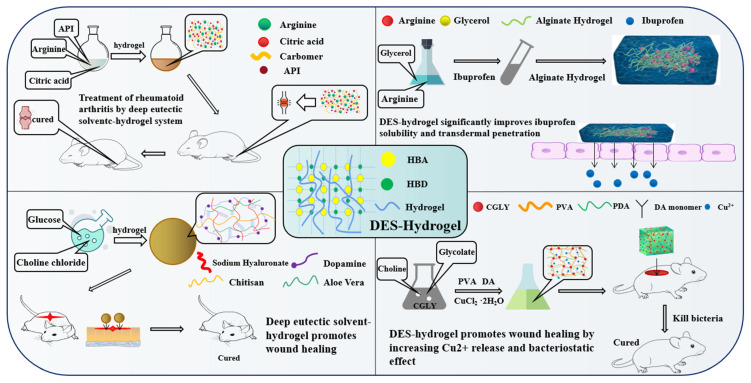
Applications of DES-hydrogel.

**Figure 5 pharmaceutics-14-02265-f005:**
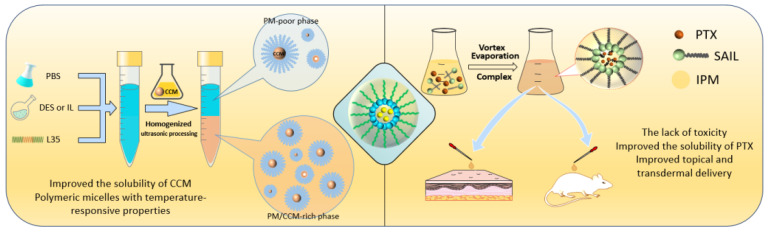
Applications of DES-micelles.

**Figure 6 pharmaceutics-14-02265-f006:**
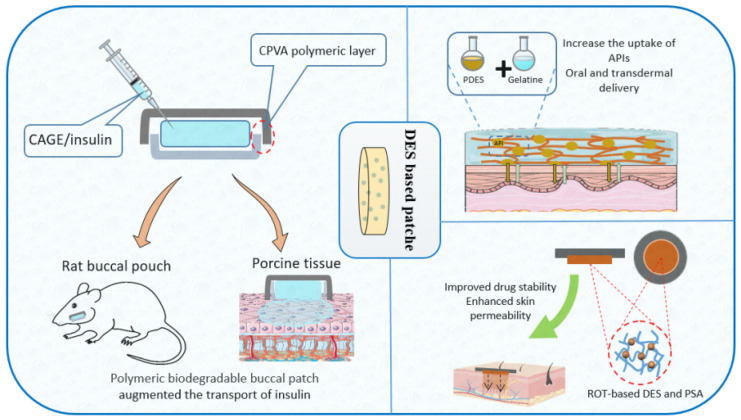
Applications of DES-patches.

**Table 1 pharmaceutics-14-02265-t001:** The composition and application of some NADESs.

HBA	HBD	Ratio	Application	References
Glucose	Lactic acid	1:5	As an efficient solvents for the extraction of phenolic	[32]
Proline	Malic acid	1:1		
Choline chloride	Sucrose	1:1/4:1		
Choline chloride	Glucose	1:1/2:1/5:2		
Choline chloride	Sorbitol	3:1/5:2		
Choline	1,2-Propanediol	1:1/1:1.5/1:2/1:3/2:1		
Choline chloride	Glucose	2:1	More efficient than methanol in extracting phenolic substances from grape skins	[34]
	Xylose	2:1		
	Glycerin	1:2		
	Fructose	1.9:1		
	Malic acid	1:1		
Glucose	Lactic acid	1:5	The stability of carthamin increased 2–8 times	[35]
Choline chloride	Sucrose	1:1/4:1		
Proline	Malic acid	1:1		
Choline chloride	Xylitol	5:2		
Choline chloride	Glucose	3:1/1:1/1:3	Increased solubility of curcumin significantly	[36]
	Fructose			
	Sorbitol			
	Xylitol			
	Maltose			
	Sucrose			
	Glycerin			
Choline chloride	Mannose	1:2	Antibacterial, anti-inflammatory, nutritional supplement	[37]
Sucrose	Citric acid	1:1	Antibacterial	[38]
Fructose, glucose	Malic acid	1:1:1		
Betaine, proline	Malic acid	1:1:1	Antioxidant activity	[39]
Choline chloride	Glucose	2:1	Anti-cancer activity	[40]
Choline chloride	Fructose			
Choline chloride	Oxalic acid	1:2	Improved extraction efficiency and conversion rate	[41]
Betaine	1,4-butanediol	1:2	Improve stability of cellulase	[42]
Choline chloride	1,4-butanediol			

**Table 2 pharmaceutics-14-02265-t002:** The composition and application of THDES.

HBA	HBD	Ratio (*w*/*w*)	Application	References
L-Menthol	Ibuprofen	(*w*/*w*) 20:80/25:75/30:70/35:65/ 40:60/50:50/60:40	Improved solubility and transdermal penetration	[43,47]
1,8-Cineole				
LD-Menthol				
Thymol				
Menthol	Ibuprofen	1:3	Development of THEDES with controlled release	[44]
Lidocaine	Ibuprofen	1:1	Improved the solubility of API	[46]
Menthol	Ibuprofen	3:1	Improved solubility and transdermal penetration	[48]
Choline chloride	Salicylic acid	1:1	Reduce waste of valuable compounds, increase the solubility of water-insoluble drugs, and overcome pharmacokinetic differences	[45]
	Aspirin	1:2		
	Paracetamol	1:1		
	Thymol	1:2		
Choline chloride	mandelic acid	1:2	Antibacterial	[49]
Atropine	Capric acid	1:2	Improved the solubility of API and avoid recrystallization	[46]
	Dodecanoic acid	1:2		
Lidocaine	Linoleic acid	1:1		
	Dodecanoic acid	1:2		
	Capric acid	1:1/1:2/1:3		
	1,8-Octanediol	1:1/1:2/1:4		
	Tetracaine	1:2		
	Prilocaine	1:3		
	Vanillin	3:2		
Osthole	Paeonol	8:2	40 times increased in solubility	[50]

**Table 4 pharmaceutics-14-02265-t004:** Preparations of DES-microemulsions and their effects.

Surfactant	Oil Phase	Water Phase	API	Application	References
Lidocaine-lauric acid (1:1/4:6)	Propylene glycol	IPM	Lidocaine	Acts as a surfactant to facilitate morphological transformation and gel formation	[67]
Choline-linoleic acid(1:1), Span-20	IPM	choline hydroxide- Propionic acid (1:1)	Insulin	Enhance the transdermal penetration of insulin through the intercellular pathway	[69]
Tween-80	DL-menthol- Acetic acid (1:2)	Water	Carotenoids	Improvement of solubility, stability and antioxidant activity of carotenoids	[133]
	DL-menthl- Octanoic acid (1:2)				
Triton X-100	Tetra-n-butylammonium chloride- n-Decanoic acid (1:2)	Water	/	Participate in the formation of green, non-toxic, readily available emulsions	[134]
Triton X-100	Tetra-n-butylammonium chloride- n-Decanoic acid (1:2)	Water	Curcumin	Improve the solubility and stability of curcumin	[127]
Tween-80, Span-20	IPM	Choline chloride-Glycerin (1:2)	Resveratrol	Improved the solubility and skin penetration of resveratrol	[128]
Span-20	IPM	Choline hydroxide-Oleic acid (1:1)	Celecoxib, Acyclovir	Increase the solubility of celecoxib and acyclovir	[129]
Cremophor EL, Glycerin	Menthol-Paeonol (6:4/5:5/4:6)	Water	Paeonol	Increased stability of paeonol and enhanced skin penetration in vitro	[130]
Tween-80	Lidocaine-Prilocaine (1:1),	Water	Lidocaine	Better stability and permeation-promoting effect on Lidocaine	[131]
Tween-80, Span-20	Lidocaine- Ibuprofen (1:2)	Water	Artemisinin	Enhanced transdermal delivery of artemisinin	[91]
/	Menthol-n-Octanoic acid (1:3)	Water	/	Preparation of pH-responsive microemulsions	[132]

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
