# Peer review of "Deep Eutectic Systems as Novel Vehicles for Assisting Drug Transdermal Delivery"

_pharmaceutics, 2022, doi:10.3390/pharmaceutics14112265_

Round 1
Reviewer 1 Report
The subject is of great interest, but the way of organization is questionable.
If you have the table of abbreviations, why you reiterate the explanations all over the body - r.12, r.27, r.30...........? Remove the table and disclose abbreviations at the first appearance.
r.35 -"...such as azone". " azone" - should start from capital letter...
There are so many of such misspellings, that it becomes hard to read.
The letter size on Figures 1,2,4 as too small.
Please reshape everything accordingly
Reviewer 2 Report
A comprehensive review on the transdermal drug delivery systems has been presented by authors in this work. Manuscript is very written and easy to understand. Tables and figures are included in the manuscript which will provide a quick summary to readers. However, following are to be addressed.
1. On Page 3, in Line 43, are “skin enhancers” skin permeability enhancers?
2. On Page 13, in Figure 1, the provided plots are very small to read.
3. On Page 14, in Figure 2, the provided images and plots are very small to read.
Round 2
Reviewer 1 Report
It looks acceptable at present form